# Sintered Glass-Ceramics, Self-Glazed Materials and Foams from Metallurgical Waste Slag

**DOI:** 10.3390/ma14092263

**Published:** 2021-04-27

**Authors:** Nicolai B. Jordanov, Ivan Georgiev, Alexander Karamanov

**Affiliations:** 1Institute for Physical Chemistry, Bulgarian Academy of Sciences, IPC–BAS, Bl. 11, Acad. G. Bonchev Str., 1113 Sofia, Bulgaria; njordanov@ipc.bas.bg; 2Institute for Information and Communication Technologies, Bulgarian Academy of Sciences, IICT–BAS, Bl. 2, Acad. G. Bonchev Str., 1113 Sofia, Bulgaria; ivan.georgiev@parallel.bas.bg

**Keywords:** foams, waste materials, sintering, hot-stage microscopy

## Abstract

The materials used for the synthesis of parent glass are 70% wt. metallurgical slag and 30% wt. industrial quartz sand. The initial batch is melted at and then quenched in water. The resulting glass frit is milled bellow 75 microns and pressed 1400 °C into “green” samples. In a next stage, they are heat treated at different temperatures with various heating rates and holding times. As a result, it is demonstrated the possibility for production variations, allowing the manufacture of three types of new materials by using the same pressed glass powders. We highlight the flexibility of the synthesis obtaining namely well densified glass-ceramics at about 950 °C, self-glazed glass-ceramics at about 1000 °C or glass-ceramic foams at approximately 1100 °C. The first set of materials is characterized by very well sintered structure combined with reasonable crystallinity; the second one—by smooth self-glazed surface with an attractive appearance and good properties and the third one—by 80–90% closed porosity and very good thermal stability above 1000 °C.

## 1. Introduction

Metallurgical companies worldwide provide waste streams in the form of different slags which can be recycled. One of the possibilities is they are mixed with the appropriate additives and vitrified. Then, after appropriate heat-treatment, the resulting glasses or frits are converted into final products such as traditional bulk glass-ceramics [1], sintered glass-ceramics [2,3] and glass-ceramic foams [4]. This approach can be considered as a smart tool for the solution of problems concerning the storage and immobilization of various inorganic wastes [5,6]. From a historical point of view, first, different materials with bulk crystallization were developed, while in recent decades, intensive research was given to the sintered glass-ceramics and foams.

Traditionally, glass-ceramic foams are prepared by the addition of inorganic carbonates or silicon carbide (SiC) [7,8] to the press powders in order for foaming to be initiated. A similar bloating process is also commonly used for the production of lightweight ceramic foam materials [9,10,11]. This can be considered as the classical way of obtaining inorganic foams.

Another case, for instance, is foaming due to the oxygen release during thermal reduction of transition metals oxides. Most often, these are iron [12] and/or manganese [13] oxides.

The latter is exactly the case presented here where it is interesting to highlight the feature of the used slag. It naturally contains iron oxides and manganese oxides simultaneously. A mechanism of auto-catalytic foam formation is realized here. In the current particular case, the formation of glass-ceramic foams is related to the release of oxygen gas as a result of the reduction of both Fe(III) oxide down to Fe(II) oxide at higher temperatures (2Fe_2_O_3_ ↔ 4FeO + O_2_↑) [14] and the partial thermal reduction of Mn(IV) oxide down to Mn(III) oxide (4MnO_2_ ↔ 2Mn_2_O_3_ + O_2_↑) [13].

The auto-catalytic bloating phenomenon performs at relatively low viscosity and temperatures above the softening point. This leads to the formation of a high amount of pores which takes place by retention of gas within the bulk [15]. However, at lower temperature, before the beginning of bloating, well sintered material can be obtained.

The aim of the presented investigation is to demonstrate the possibilities for production of various glass-ceramic products from waste raw materials with different applications within a single production line using the same initial glass powder by application of different firing regimes. In this way, various materials, with different structure depending on the heating temperatures and rates, can be obtained.

## 2. Materials and Methods

An iron and manganese containing slag from the iron and steel company Helwan in Cairo Governorate, Eqypt was used for the thermal synthesis of the investigated glass-ceramic samples. Due to the low amount of main glass-formers (SiO_2_ + Al_2_O_3_ ≈ 35% wt.) in the slag, 30% wt. industrial quartz sand were added to 70% wt. slag. Thus, the percentage of the glass formers reached about 55% wt., which guarantees good chemical durability together with a moderate melting temperature of 1400 °C. It can be noted that the presence of high amounts of amorphous phase in the slag also favorizes the melting. The exact procedure concerning the glass melting was explained in our previous work [16].

After 2 h of holding, the melt was quenched in water. The resulting glassy frit was characterized with the following chemical composition in % wt. determined by XRF measurement: 49.2 SiO_2_; 5.1 Al_2_O_3_; 5.5 Fe_2_O_3_; 18.6 CaO; 1.1 MgO; 5.8 MnO; 10.9 BaO; 0.7 TiO_2_; 0.7 K_2_O [16]. It is interesting to be pointed out the result that the presence of Fe and Mn in the slag is unusually high.

Thus, obtained frit was grinded with a planetary mill FRITSCH (Germany). Portions of 5 g each were then sieved to a fraction below 75 microns with a digitally programmed sieving machine CISA (Spain). The sieving times were programmed to10 min. Finally, the “green” samples with a parallelepiped shape of 50 × 5 × 4 mm were prepared by uniaxial pressing of the powders at40 MPa with a pneumatic pressing machine NANNETTI (Italy). Then, 7% wt. polyvinyl alcohol (PVA) was used as a binding agent as well.

In all investigations, we used a thermo-optical computerized system ESS ODLT HSM-1400 MISURA (Italy) allowing measurements with heating rates up to 30 °C min^−1^. This equipment combines two measurement techniques: horizontal contact-less optical dilatometry and vertical hot-stage microscopy (abbreviated below as ODLT and HSM as already mentioned). Both methods trace the length and height change, respectively, of the samples with different sensitivity and can be switched and used on demand. This turned out to be an established laboratory method in recent years, since it is reliable and fast and is used already by many research groups worldwide [9,10,11,17].

Thus, obtained samples are appropriate for studies with ODLT. Some of these samples were cut to (6–8) × 5 × 4 mm pieces for additional thermal treatments at higher temperatures.

In particular, ODLT was used toward the investigations of the process of sintering while HSM was used for the sake of a general overall investigation of the high temperature thermal treatment behavior and for tracking of the foaming process.

Other green smaller samples, suitable for direct HSM measurements, with standard cylindrical shape of 5 mm height and 2 mm diameter, were prepared by manually pressing with a manual plunger and distilled water as a binder.

The following types of measurements were performed for the purpose of process optimization by means of adjusting thermal scan rates, firing temperatures and for a detailed investigation of the studied processes: 1. Non-isothermal and isothermal HSM scanning of the small manually pressed samples; 2. Isothermal ODLT measurements in the temperature range of 900–950 °C using 50 mm samples; 3. Preparation of self-glazed and foamed cut samples at short holding times and higher temperatures.

Scanning electron microscopy (SEM) images of surfaces and fractures of the sintered glass-ceramics were taken with a JEOL 6390 (Japan) microscope. Following the laboratory procedure proposed by Strnad [18], the bulk samples were polished and etched 5 s each in a 2% wt. aqueous solution of hydrofluoric acid (HF).

The phase composition of the powdered sintered glass-ceramics was determined by X-ray diffraction spectroscopy (XRD) with a PANALYTICAL EMPYREAN (USA) spectrometer. 

Finally, the entire bulk structure of the newly formed sintered material was studied by 3D X-ray tomographic analysis. In this case, industrial computed tomographic imaging (µCT) was carried out with a Nikon XT H 225 system, developed by Nikon Metrology. Each sample was rotated on 360° and 2520 digital radiographic projections were acquired during the rotation. The Inspect-X CT software was used to control the acquisition process. The scanning of the samples was performed with a voltage of 110 kV, 100 µA tube current and exposure time of 500 ms. The size of the voxel was 4 µm. The volume reconstructions were performed by CT Pro 3D software developed by the producer of the equipment. Volume rendering, porosity analysis and measurements were performed by VG Studio Max 2.2 software by Volume Graphics Inc. (Heidelberg, Germany)

When we have foams of irregular structure, arbitrary shape and broad cell volume distribution the µCT scanning turns out to be the most proper method toward the non-destructive characterization of cellular materials [19,20].

## 3. Results and Discussion

In Figure 1, is presented a typical HSM curve of the variation of the height of a single press-powder sample in the domain thermal expansion—temperature.

Additionally, in Figure 1a–e, characteristic snapshots are indicated during the thermal cycle presented in Figure 1. In this manner, the possibility to obtain different materials is demonstrated. In Figure 1a, the thermal propagation is shown with a snapshot at 700 °C, which is practically identical with the initial sample. The densification process starts at about 800 °C and the sintering of the glass-ceramics is completed at about 900 °C as it is shown in Figure 1b. Further scanning up to 1050 °C led to no serious change in the volume of the sample. At this temperature, however, some certain smoothing of the shape is observed and after that, initiates the beginning of the foaming process. Further, Figure 1c corresponds to material formation characterized with smooth auto-glazed surface due to the fact that this temperature is slightly above the HSM softening point [21,22].

In Figure 1d, the shape is demonstrated at 1150 °C, which corresponds to maximal extent of bloating during foaming. Further increase of the temperature up to 1200 °C as indicated in Figure 1e leads to a rapid structural shrinkage and collapse of the foam. The sample reveals a nearly spherical silhouette (c.f. Figure 1e) which means that at this temperature, the gas release is completed and the apparent viscosity decreases due to the melting of the formed at heating crystal phases. This assumption was established with DTA in our previous work [16].

It can be highlighted that the HSM measurements reveal three well distinguished temperature intervals for synthesis of three different types of new materials.

At low temperature, before the beginning of the foaming sintered glass-ceramics was investigated.

The densification and the crystallization are two major competing processes during thermal treatment of sintered glass-ceramic products. Both should be balanced very carefully in tuning the synthesis since one should consider the close temperature interval where both processes are taking place.

The sintering ability of glass powders is a key feature characterizing the manufacturing of these glass-ceramics. In principle, it is favored by using finer powder fractions and/or the implementation of higher heating scan rates. In this manner, the inhibiting effect of the crystallization on the densification can be reduced.

The second key factor is the crystallization process. It is well known that if the composition possesses a low crystallization ability, the densification can be explained with the theory of viscous flow sintering. On the contrary, in the case of higher crystallization trend, the latter can partially inhibit the sintering.

At the same time, the firing regimes can guarantee the completion of the crystallization process. The resulting precipitation is higher at lower crystallization temperature, which can lead to bigger percentage of formed crystal phases [5,23]. However, at lower temperatures, the crystallization times are longer.

For instance, the relationship between densification and new phase formation was elucidated in our previous work [16]. We investigated the effect of CaF_2_ on the sintering ability and the foaming trend of the same base glass studied there (with different additions of CaF_2_). In small quantities, due to viscosity diminishing, CaF_2_ decreases the sintering and the melting temperatures, while in larger quantities, it blocks the sintering and inhibits the foaming of the sample.

With the increase of the temperature in the composition without CaF_2_, which is the case under discussion, deformation starts, followed by bloating, which is related to the presence of iron and manganese oxides in the parent glass. This gives the possibility of foam glass-ceramics to be studied.

The mechanism of bloating by Fe(III) oxide reduction is well known. Sandrolini and Palmonary, 1976 [9], provided a detailed description of the key role of iron oxides in the foaming of vitrified sintered ceramic materials. At elevated temperatures, Fe_2_O_3_ is partially reduced with the release of oxygen as the bloating gaseous phase. The latter generates large pores within the fired body and determines a density decrease [15].

The auto-catalytic foaming can be explained with the variation of the ratio of the oxidation state of the transition metal oxides with temperature. The higher the temperature, the lower the amount of the reduced form.

It can be mentioned that during quenching of the melt into frit, an equilibrium ratio of the redox couple Fe(II) ↔ Fe(III) was “frozen” (fixed) in the glass. In this case, it corresponds to the equilibrium value of the melting temperature.

However, above the glass transition temperature, oxidation takes place. This practically leads to complete oxidation of Fe(II) into Fe(III) [24,25]. With further temperature increase, the reduction initiates again. It is interesting to note that when the heat treatment is carried out in an inert atmosphere, no oxidation and subsequent reduction related with foaming is observed [14].

So that with temperature variation and with the related viscosity and oxi-reduction processes, the production of different sintered glass-ceramic materials is possible.

### 3.1. Synthesis and Characterization of Sintered Glass-Ceramic Materials

The low temperature sintering behavior of a studied glass-ceramic sample is presented in Figure 2. Results are obtained at the isothermal heat treatment step at the optimal temperature of 950 °C and with a holding time of 30 min using a modest heating rate of 10 °C min^−1^. The densification interval is indicated by the shaded area. This result highlights that the sample shrinks by 13% and that the sintering process is entirely completed already in the beginning of the non-isothermal step (we have 12% linear shrinkage during heating and only 1% taking place during the first 5 min of the hold). It is important to note that during the remaining 25 min, no volume changes have been observed, i.e., no deformation is carried out during the crystallization step. We have to mention additionally that at lower temperatures [5,15,23], longer crystallization times can be applied.

The complete crystallization process for 30 min at 950 °C has been confirmed by XRD measurements. In Figure 3, are presented XRD diffraction patterns of the initial glass (3a) and sample sinter crystallized at 950 °C for 1 min (3b) and for 30 min (3c). It is evident that during the isothermal step, some increase of crystallinity is observed and the main crystal phase is pyroxene. By comparison of the intensities of the amorphous halo of the initial glass with that of the crystallized sample for 30 min was found that the crystallinity is 35 ± 5% [19].

From observations of the SEM image shown in Figure 4a, one can estimate that the porosity in the bulk of the sintered new material is mainly of closed type and its amount is less than 10% vol. A typical spherical pore is presented in Figure 4b, while Figure 4c reveals a sharp-edge inter-granular porosity with visible channels between the grains. The surface is smooth and very well sintered as demonstrated in our previous recent study [14]. Nevertheless, some closed porosity with a structure similar to Figure 4c is elucidated (c.f. Figure 4d). The presence of such type of porosity is probably indicative for certain overlapping of both sintering and crystallization processes as it was highlighted by the isothermal sintering and XRD results.

### 3.2. Synthesis and Characterization of Self-Glazed Glass-Ceramic Materials

Figure 1 indicates that after non-isothermal heat treatment, the sintering curve up to 1000–1050 °C reveals no significant volume change. At the same time, the comparison between the silhouette snapshots in Figure 1b,c shows some smoothing of the surface due to the viscosity decrease.

Thus, the possibility to form self-glazed glass-ceramic samples was studied. If we terminate the firing at the above cited temperature interval, a well densified material with smooth auto-catalytically induced (or self-glazed) surface can be successfully obtained. This densification is enhanced by applying higher heating rates which gives a possibility to shorten the firing regime.

In Figure 5, is shown a representative species obtained at 1020 °C using a scan rate of 20 °C min^−1^ and holding time of 3 min [16].

This material is characterized with zero water absorption, about 15% vol. closed porosity and good mechanical properties (apparent density of 2.22 ± 0.05 g cm^−3^ and hardness of 4.71 ± 0.01 GPa) and compressive strength of 140 ± 20 MPa [16]. These properties correspond to natural stones and some construction materials.

Working at higher temperatures and/or higher heating rates, due to the formation of lower amount of crystal phase and decreasing of the apparent viscosity some deformation (related in this particular case to auto-glazing and color change) is observed. This material is characterized with a commercial shiny appearance appropriate for e.g., decorative swimming pool tiling.

### 3.3. Synthesis and Characterization of Glass-Ceramic Foam Materials

The production of low weight fire resisting foams is the third and final part of the current research. This is directly realized as a result due to the presence of iron oxides (FeO and Fe_2_O_3_) and manganese oxides (Mn_2_O_3_ and MnO_2_) in the initial glass frit.

As it was already pointed out, after almost completed oxidation of the transition metals in the glass transition region with the further increase of the temperature thermal reduction of iron and manganese starts again.

Moreover, the already reached degree of sintering is directly responsible for the subsequent foaming trend of the glass-ceramics as well, in a fashion such that, higher initial degree of sintering leads to a higher foaming ability and vice versa. This circumstance was confirmed in our previous work [14] and by other authors [12].

As a result, in well sintered glass-ceramics with closed porosity and with moderate crystallinity, which is the case of discussion, the possible structural expansion with the increase of temperature is very intensive. We highlighted also that carrying out synthesis with higher heating rates of e.g., 30 °C min^−1^ leads to lower crystallinity and thus to lower apparent viscosity. Despite the shorter heat treatment times, the beginning of the foaming process at 30 °C min^−1^ starts at lower temperature compared to that at 5 °C min^−1^ [16]. Thus, in the production of the foam, compared to the synthesis of sintered glass-ceramics at 950 °C, the maximal heating rate of 30 °C min^−1^ is used.

As it is shown in Figure 1, the intensive increase of the volume starts taking place at a temperature of about 1050 °C and the maximal expansion is reached at about a 100 °C higher temperature.

The appropriate regime was estimated in a series of preliminary measurements. An optimal isothermal HSM measurement with holding time of 30 min at 1100 °C of the formation of new glass-ceramic foam material sample ending up with its stabilization is shown in Figure 6. The snapshot selection of three characteristic pictures in Figure 6 recorded during the initial sintered state of the sample (6a), the maximum height of bloating after about 10 min, of holding (6b) and the stabilization of the shape after about 20 min of a glass-ceramic foam (6c) are presented here. The observed stabilization can be explained most probably with the completion of the evolution of oxygen release. After the fixing of the shape, no change of the sample is observed at cooling. This elucidates that the material can be considered as fire resistant at temperatures below the range of 1000–1050 °C.

A larger foamed sample obtained under the same optimal conditions is represented in Figure 7d together with: 7a initial non-sintered sample; 7b sintered at 950 °C at 10 °C min^−1^ and 7c, self-glazed sample. All the samples were prepared by cutting of initial bars as described in the Experiment section. The comparison between sample 7b and 7d shows a huge linear expansion of about four times and color variation.

The same samples were investigated further with X-ray computed tomography (µCT). In the observations, were evident sub-resolution effects as well, contributions that cannot be clearly seen. Thus, the information obtained is correct only for objects larger than about 10 µm. The initial sample in Figure 8a is abundant with open pores of irregular shape with size up to 60–80 µm, i.e., their size is comparable to this of the initial particles. In Figure 8b,c are presented sintered glass-ceramics and sintered self-glazed samples, respectively. In both samples, the porosity is of closed type and mainly with spherical shape. However, in the second object, the maximal size of the porosity increases more than three times, reaching 80–100 µm. This indicates the beginning of the coalescence process which is evident mainly in the middle of the sample. Figure 8d demonstrates the image of the boundary between the most dense surface layer and the volume part where the coalescence is highlighted.

Figure 9 shows the structure of the foamed sample. In 9a, is given a 3D volume reconstruction of the whole sample, while in Figure 9b,c are shown 2D images of slices from the center part of the objects at different magnifications.

In Figure 9a as well as in Figure 7d, is elucidated that the specimens’ surface is smooth and practically without open pores. In fact, this confirms, as indicated in our previous work [16], that the similar species does not absorb water even after 48h of sinking.

The distribution and the structure of the porosity are highlighted in Figure 9b,c. We may note that there are two main types of pores: primary large pores and secondary fine pores abundant in the walls of the foam. The structure of the foam is cellular with a nearly spherical porosity. The primary pores are sized between 500 and 2000 µm, while the secondary are mainly of sizes between 50 and 100 µm.

Probably the formation of secondary pores is related to a more complex reduction process involving iron and manganese reduction. In fact, in traditional ceramics and glass-ceramics foams, such an intriguing structure was not observed [26,27,28].

The 3D reconstruction of the sample also gives the possibility to evaluate the total porosity. The measurement of the volumes of the whole sample and that of the total solid substance produces porosity percentage in the range of 85–90% vol., which is a very reasonable value.

## 4. Conclusions

In the present investigation, we demonstrated the possibility for carrying out successful synthesis of different sintered materials obtained from the same glass, products of vitrification of metallurgical slag. By variation of the used firing regime, generally speaking, we discuss the possibility for manufacturing the following three types of materials: 1. sintered glass-ceramic; 2. self-glazed material; 3. glass-ceramic foam. 

The demonstrated technological flexibility is controlled by the heat treatment rates and holding temperatures because of the variations of the crystallinity and of the apparent viscosity, as well as by the bloating tendency due to the thermal reduction of iron and manganese oxides present in the used slag.

By using a heating rate of 10 °C min^−1^ and heat treatment of 950 °C, we obtain well sintered glass-ceramics characterized with about 10% vol. porosity and moderate crystallinity of about 35%.

Applying higher rates of 20 °C min^−1^ and short holding times at 1020 °C, we obtain a product with slightly misshapen surface in the form of self-glazing with very pleasant to the eye smooth surface.

Finally, at thermal scan rates of e.g., 30 °C min^−1^ and holding at temperatures of 1100 °C, allows us to obtain new fire-resisting sintered glass-ceramic foams with cellular structure and closed porosity of about 85–90%.

## Figures and Tables

**Figure 1 materials-14-02263-f001:**
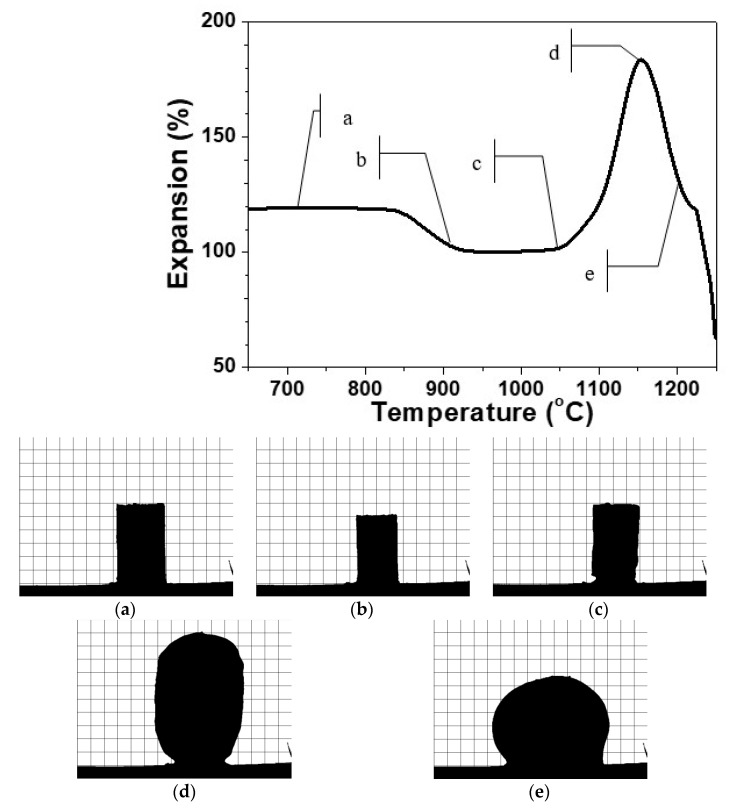
HSM curve of glass ceramic sample at 10 °C min^−1^. Figure 1a–e: profile optical images of the glass-ceramics at the respective temperature of 700°C (**a**), 900 °C (**b**), 1050°C (**c**), 1150 °C (**d**) and 1200 °C (**e**).

**Figure 2 materials-14-02263-f002:**
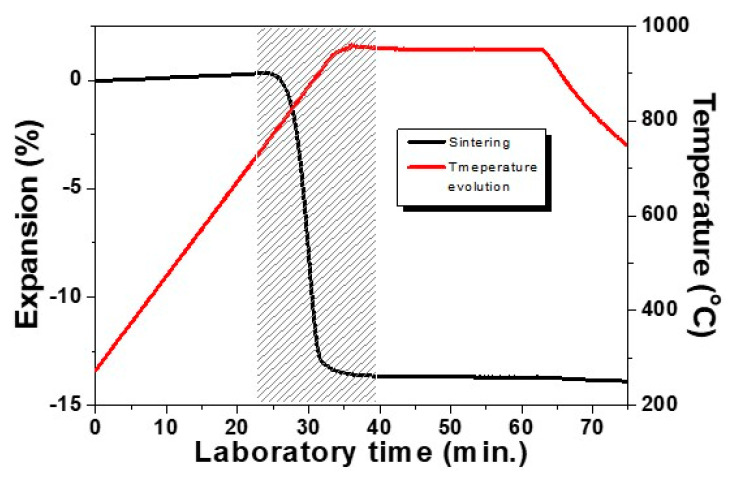
Sintering curve of glass–ceramic sample revealed by HSM.

**Figure 3 materials-14-02263-f003:**
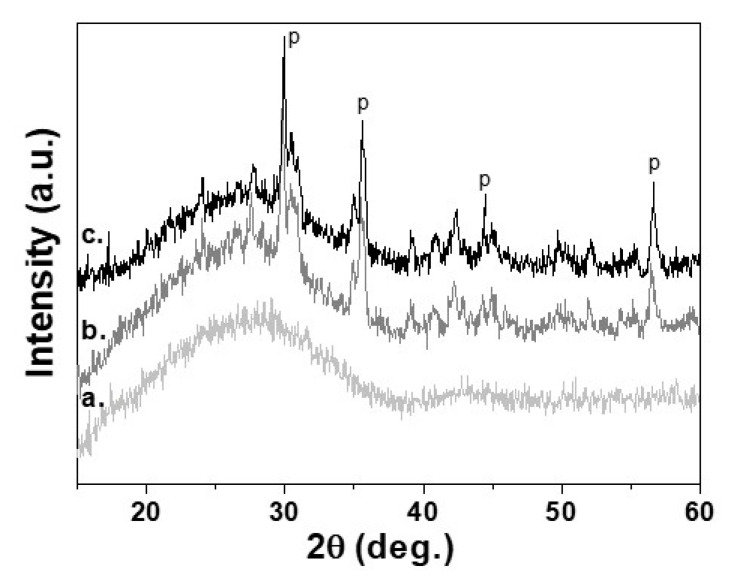
XRD patterns of a. Parent glass, b. Glass-ceramics isothermally sintered at 950 °C for 1 min; c. Glass-ceramics isothermally sintered at 950 °C for 30 min. “P” referes to the peaks of pyroxene crystal phase.

**Figure 4 materials-14-02263-f004:**
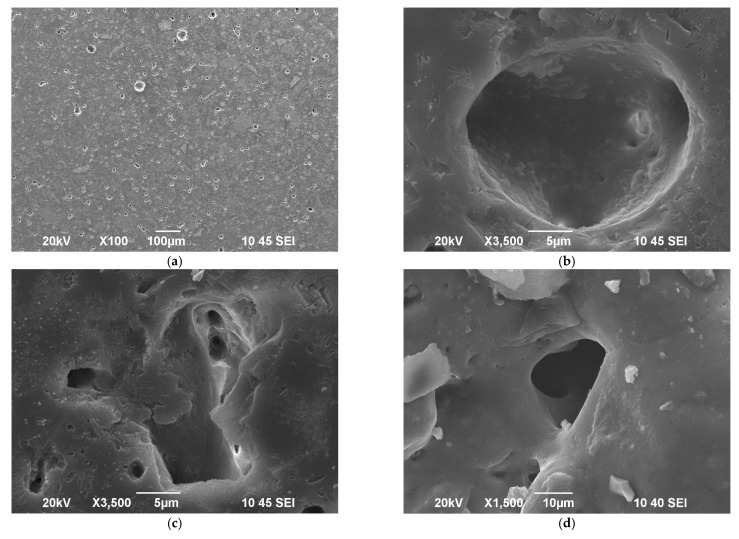
SEM images of polished samples at different magnifications: (**a**–**c**); Single pore formed on the surface, (**d**).

**Figure 5 materials-14-02263-f005:**
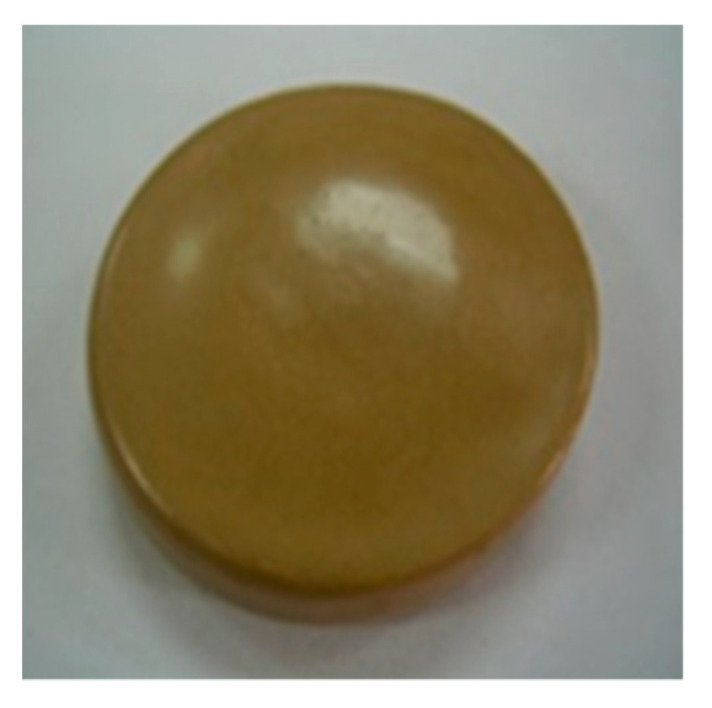
Color photograph of bloated iron-rich glass-ceramic sample as described in [14].

**Figure 6 materials-14-02263-f006:**
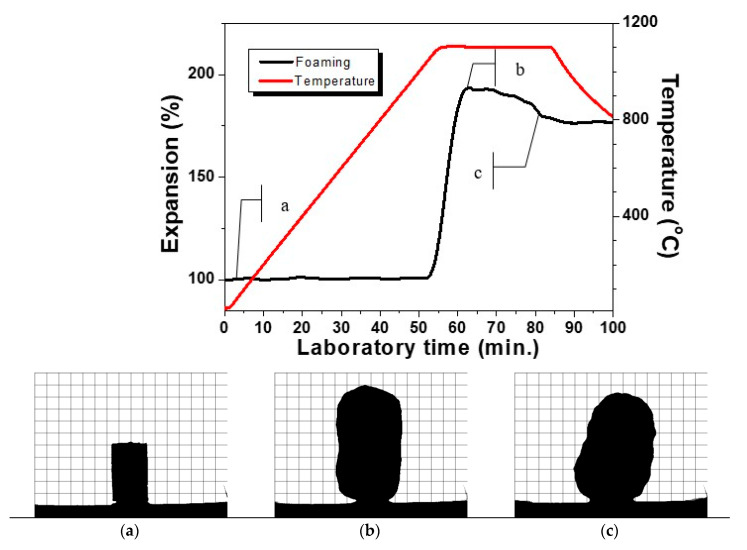
HSM sintering curve during the formation of a glass ceramic foam at a temperature of 1100 °C; (**a**–**c**)—silhouette images during the bloating process.

**Figure 7 materials-14-02263-f007:**
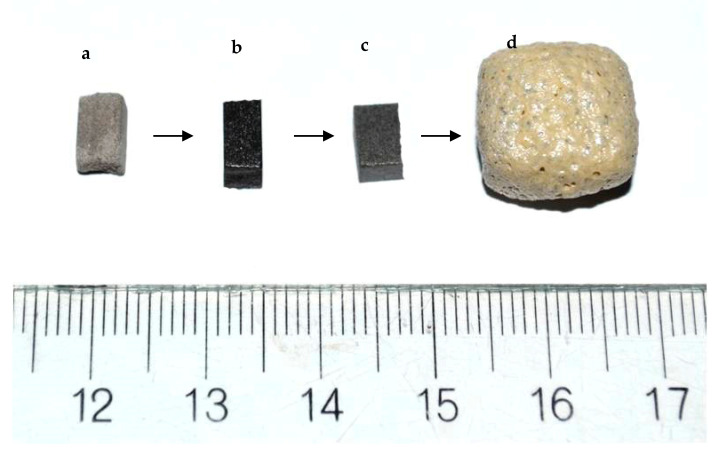
Photographs of the studied materials: (**a**) Press-powder; (**b**) Sintered glass-ceramics synthesized at 900 °C; (**c**) Self-glazed glass-ceramics obtained at 1020 °C; (**d**) Glass-ceramic foam sample bloated at 1100 °C.

**Figure 8 materials-14-02263-f008:**
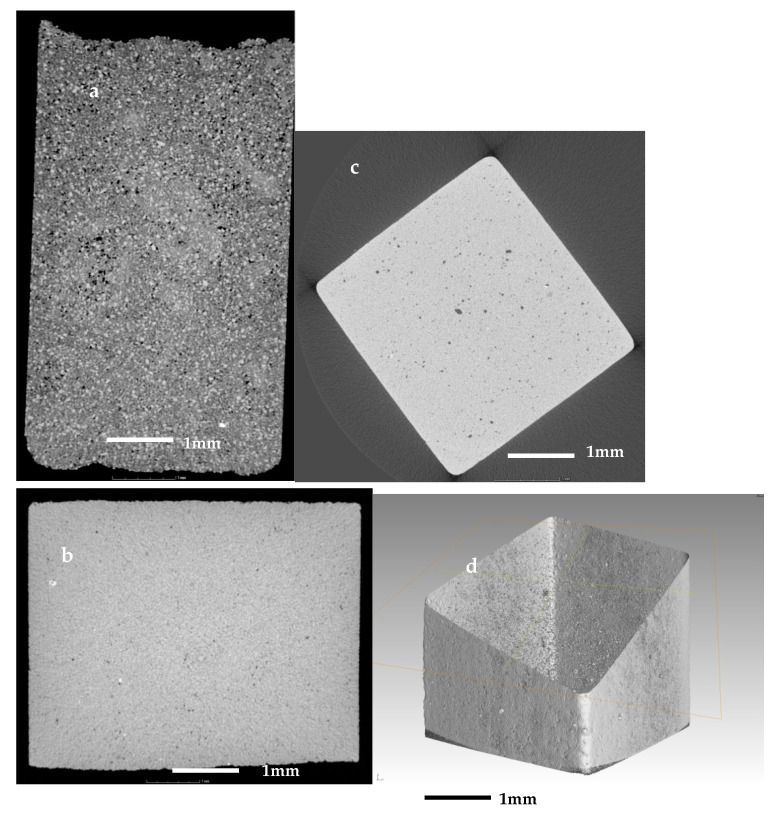
X-ray CT scan images of (**a**) the initial sample; (**b**) of a sintered glass-ceramics; (**c**,**d**) sintered self-glazed sample.

**Figure 9 materials-14-02263-f009:**
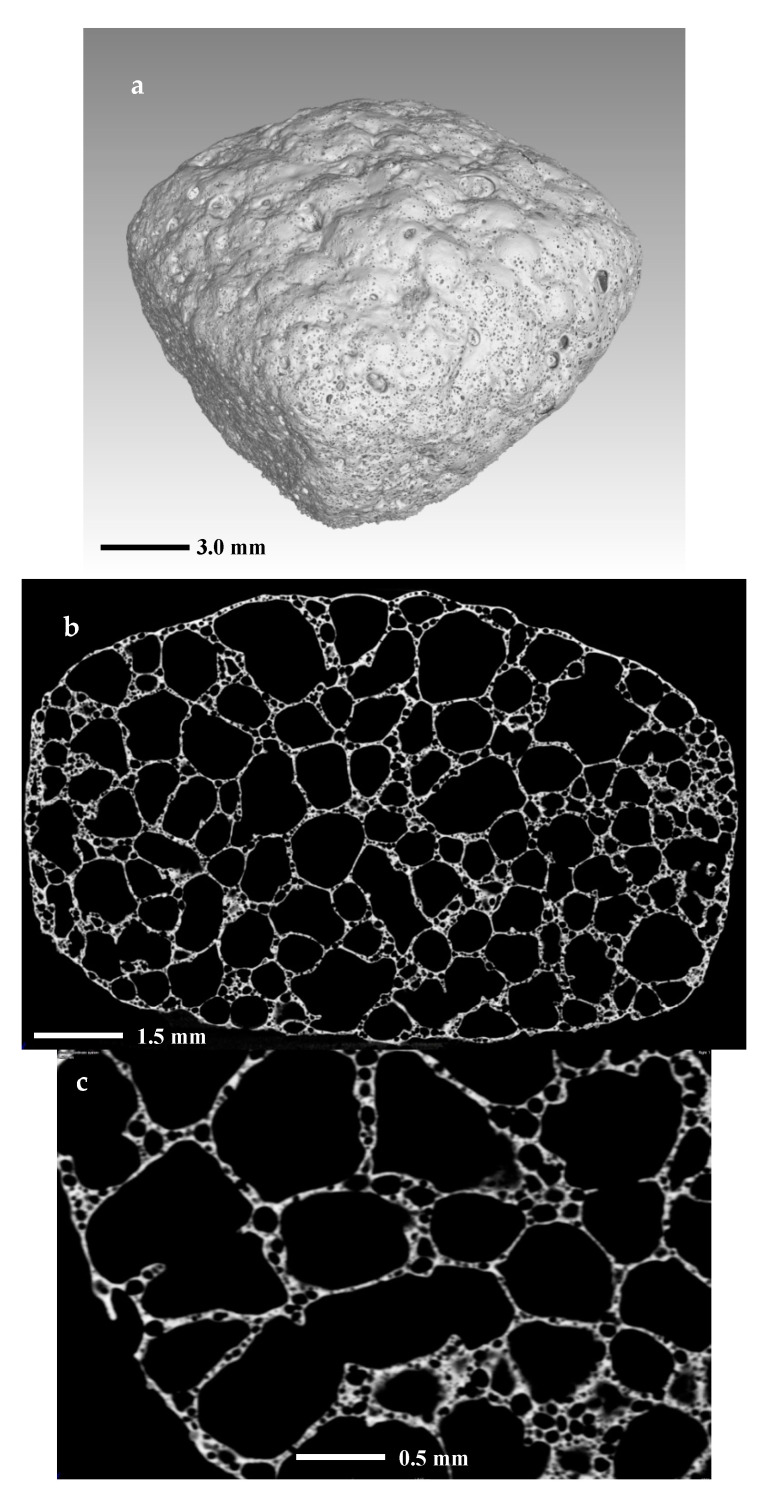
X-ray CT scan images of (**a**) 3D reconstruction of foam sample; (**b**,**c**) 2D intersection.

## Data Availability

https://www.intechopen.com/books/foams-emerging-technologies/sintered-iron-rich-glass-ceramics-and-foams-obtained-in-air-and-argon (accessed on 27 April 2021).

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
