# Peer review of "Sintered Glass-Ceramics, Self-Glazed Materials and Foams from Metallurgical Waste Slag"

_materials, 2021, doi:10.3390/ma14092263_

Round 1

Reviewer 1 Report

The manuscript provides interesting results on the production of different types of materials from the metallurgical slag-quartz powders mixture as a function of different heat treatment. This topic is up-to-date and the presented results might be of high interest to the materials science community. On the other hand, the paper is quite difficult to read and follow.

To me, the paper can be accepted for publication in Materials provided the following comments are addressed by the authors:

1. The presentation and discussion of the result are rather poor and need proper revision in order to enhance the “flow” of the paper. There are also numerous grammar issues, typos and misspellings. The manuscript should be proof-read by a native speaker. In addition, the results should be discussed in more detail with respect to the available literature.

2. Lines 49-74: These paragraphs should be better utilized in the Discussion section

3. Lines 139-166: These paragraphs should appear in the Introduction or later in the Discussion section rather than in the beginning of the Results section.

4. Scale bars are missing in the majority of Figures where applicable.

5. Subscripts should be used in the chemical formulae, e.g. line 55: Fe2O3 -> Fe2O3

6. Figure 8c: A zoomed-in micrograph showing the glaze/interior interface should be shown as evidence of glaze thickness and quality.

Further minor comments:

Lines 47-48: Entire chemical reactions should be provided.

Lines 81-84: What is the origin of the slag? Even though you refer to your previous work, some basic information about the material should be provided. Also, please elaborate more on the motivation/effect of quartz addition and why the fraction of 30 wt.% was selected.

Lines 85-86: How was the composition determined?

Line 97: „The latter combines...“ From the description, the thermo-optical computerized system ESS ODLT 96 HSM-1400 MISURA mentioned in the text seems to be a single device – this is confusing; please elaborate on differences between ODLT and HSM.

Lines 124-133: What was the resolution of the 3D X-Ray tomography (voxel size)?

Lines 149-152: what do you mean by „crystallization trend”? Maybe crystallization rate?

Figure 1: Please provide info about the heating rate.

Suggestion: Mechanical properties (hardness, strength, energy absorption, etc.) should be determined in order to assess the sample integrity (i.e. quality of the initial powder compaction after sintering at different conditions) – may be an objective of ongoing/future work.

Author Response

  1. The presentation and discussion of the result are rather poor and need proper revision in order to enhance the “flow” of the paper. There are also numerous grammar issues, typos and misspellings. The manuscript should be proof-read by a native speaker. In addition, the results should be discussed in more detail with respect to the available literature.

A: the text was carefully checked and many mistakes were corrected. Additionally stylish and grammatical modifications were made as well.

  1. Lines 49-74: These paragraphs should be better utilized in the Discussion section

A: The part of lines 49-52 has been modified, while lines 53-74 have been replaced after Figure 1.

  1. Lines 139-166: These paragraphs should appear in the Introduction or later in the Discussion section rather than in the beginning of the Results section.

            A: The text has been moved and modified.

  1. Scale bars are missing in the majority of Figures where applicable.

            A: Appropriate scale bars have been added accordingly in Figs. 5, 8 and 9.

  1. Subscripts should be used in the chemical formulae, e.g. line 55: Fe2O3 -> Fe2O3

A: The numbers in the chemical formulae have been subscripted on line 55 and all over the text.

  1. Figure 8c: A zoomed-in micrograph showing the glaze/interior interface should be shown as evidence of glaze thickness and quality.

            A: Additional Figure 8d with relevant comments was added. 

Further minor comments:

Lines 47-48: Entire chemical reactions should be provided.

            A: Chemical reactions have been listed in brackets in the same lines.

Lines 81-84: What is the origin of the slag? Even though you refer to your previous work, some basic information about the material should be provided. Also, please elaborate more on the motivation/effect of quartz addition and why the fraction of 30 wt.% was selected.

            A: Information has been given in the text in the same paragraph.

Lines 85-86: How was the composition determined?

            A: Answer has been given on the same lines in the text.

Line 97: „The latter combines...“ From the description, the thermo-optical computerized system ESS ODLT 96 HSM-1400 MISURA mentioned in the text seems to be a single device – this is confusing; please elaborate on differences between ODLT and HSM.

A: Additional explanation of the measurement techniques has been added in the same paragraph.

Lines 124-133: What was the resolution of the 3D X-Ray tomography (voxel size)?

            A: An answer is given in Experimental.

Lines 149-152: what do you mean by „crystallization trend”? Maybe crystallization rate?

A: We have changed the term “trend” with the more appropriate word “ability”. From our point of view, the crystallization rate can be used if we have information about the real crystal growth rate in micrometers per minute.

Figure 1: Please provide info about the heating rate.

            A: The heating rates have been given in the caption of Figure 1 in the text.

Suggestion: Mechanical properties (hardness, strength, energy absorption, etc.) should be determined in order to assess the sample integrity (i.e. quality of the initial powder compaction after sintering at different conditions) – may be an objective of ongoing/future work.

A: Unfortunately, at the moment we don’t have enough data of the mechanical properties. However, we are planning the obtaining of appropriate equipment for measurements of these characteristics.

Reviewer 2 Report

Reviewed paper presents interesting and important aspect of recycling of metallurgical wastes. Presented studies show promising effects of obtaining different types of ceramic materials and, in my opinion, deserve to be published. However text needs polishing, e.g.:

28 – …glasses or frit – s is missing for plural

55 – subscript in Fe2O3 is missing

155 – timesare

208 – itysis30

And many other editorial mistakes.

I suggest to shorten the introduction part, because it is a bit too long.

I also have substantive questions/suggestions:

What was the slag composition? It is a very important information from the technological point of view. In order to be able to use such waste, its chemical composition must be known. I would suggest running XRD, XRF and FT-IR measurements.

Why were the samples etched in HF?

Due to amorphous halo visible in XRD patterns it is highly recommended to conduct XRF, FT-IR or Raman studies. The composition of final product determines its later potential use and thus increases the interest of the reader/industry.

Author Response

28 – …glasses or frit – s is missing for plural

A: According to our knowledge the word “frit” has only singular form when we discuss glass-powders with identical chemical composition. Plural can be used with different chemical compositions.

55 – subscript in Fe2O3 is missing

            A: Subscrits have been added.

155 – timesare

A: The error is corrected.

208 – itysis30

            A: The error is corrected.

And many other editorial mistakes.

            A: The text has been entirely read and checked.

I suggest to shorten the introduction part, because it is a bit too long.

            A: The Introduction has been modified according to the suggestion of both reviewers.

I also have substantive questions/suggestions:

What was the slag composition? It is a very important information from the technological point of view. In order to be able to use such waste, its chemical composition must be known. I would suggest running XRD, XRF and FT-IR measurements.

A: The chemical and phase composition of the slag was already reported in our previous work [16]. Here for the sake of clarification we repeat the information for the low amount of glass-formers in the slag which is mainly in the glassy form (Due to the low amount of main glass-formers (SiO2 + Al2O3 » 35 % wt.) in the slag, 30 % wt. industrial quartz sand have been added to 70 wt. % slag. Thus the percentage of the glass formers reaches about 55 % wt., which guarantees good chemical durability together with a moderate melting temperature of 1400 oC. It can be noted that the presence of high amounts of amorphous phase in the slag also favorizes the melting. The exact procedure concerning the glass melting was explained in our previous work [16].)

Why were the samples etched in HF?

            A: We have followed the standard experimental procedure given e.g. by Strnad [18].

Due to amorphous halo visible in XRD patterns it is highly recommended to conduct XRF, FT-IR or Raman studies. The composition of final product determines its later potential use and thus increases the interest of the reader/industry.

A: From the point of view of the evaluation of the crystal phase formed we have used the decrease of the intensity of the amorphous halo as a function of the crystallinity. This approach gives acceptable results at low and moderate crystallinity of the sample which is the case of current discussion. We agree with the comment about the Raman spectroscopy but unfortunately we currently don’t have an access to such instrument.

Round 2

Reviewer 2 Report

Thank you for the answers. The paper, in my opinion, is ready for publishing.